# Combinatorial multimer staining and spectral flow cytometry facilitate quantification and characterization of polysaccharide-specific B cell immunity

Dennis Hoving [1✉], Alexandre H. C. Marques[1], Wesley Huisman [1], Beckley A. Nosoh [1], Alicia C. de Kroon[1], Oscar R. J. van Hengel [1], Bing-Ru Wu[1], Rosanne A. M. Steenbergen[1], Pauline M. van Helden[2], Britta C. Urban[3,4], Nisha Dhar[5], Daniela M. Ferreira [3,4], Gaurav Kwatra[5,6], Cornelis H. Hokke [1] & Simon P. Jochems [1✉]

Bacterial capsular polysaccharides are important vaccine immunogens. However, the study of polysaccharide-specific immune responses has been hindered by technical restrictions. Here, we developed and validated a high-throughput method to analyse antigen-specific B cells using combinatorial staining with fluorescently-labelled capsular polysaccharide multimers. Concurrent staining of 25 cellular markers further enables the in-depth characterization of polysaccharide-specific cells. We used this assay to simultaneously analyse 14 *Streptococcus pneumoniae* or 5 *Streptococcus agalactiae* serotype-specific B cell populations. The phenotype of polysaccharide-specific B cells was associated with serotype specificity, vaccination history and donor population. For example, we observed a link between non-class switched (IgM+) memory B cells and vaccine-inefficient *S. pneumoniae* serotypes 1 and 3. Moreover, B cells had increased activation in donors from South Africa, which has high-incidence of *S. agalactiae* invasive disease, compared to Dutch donors. This assay allows for the characterization of heterogeneity in B cell immunity that may underlie immunization efficacy.

[1] Leiden University Center for Infectious Diseases, Leiden University Medical Center, Leiden, The Netherlands. [2] AIMM Therapeutics, Amsterdam, The Netherlands. [3] Department of Clinical Sciences, Liverpool School of Tropical Medicine, Liverpool, UK. [4] Oxford Vaccine Group, University of Oxford, Oxford, UK. [5] Vaccines & Infectious Diseases Analytics, University of Witwatersrand, Johannesburg, South Africa. [6] Department of Clinical Microbiology, Christian Medical College, Vellore, India. ✉email: d.hoving@lumc.nl; s.p.jochems@lumc.nl

Capsular polysaccharides (PSs) form a significant component of the external cellular surface of multiple microbial species. These carbohydrate chains portray extensive biochemical diversity coming from the constituting monosaccharides and their type of linkage, which results in distinct serotypes with varying antigenicity, immunogenicity and pathogenicity[1]. Serotype-specific PS are immunodominant antigens and commonly used as vaccine targets to induce protective B cell responses and neutralizing antibodies, as shown by current vaccines against *S. pneumoniae* (Spn or pneumococcus), *Haemophilus influenzae* type b, *Neisseria meningitidis* and *Salmonella typhoid* VI[2].

Capsular PS-conjugate vaccines have significantly reduced the global burden of disease; however, further development is imperative. For example, despite the licensing of the first pneumococcal vaccine in the year 1977, Spn remains the principal etiologic bacterial agent of pneumonia, claiming approximately 300,000 infant deaths per year globally[3]. This high mortality is largely due to serotype-associated efficacy of vaccination, coverage, replacement, geographic distribution and invasiveness[4–9]. Consequently, multiple novel pneumococcal PS vaccine formulations are being developed[4]. In addition to Spn, vaccine development of currently non-covered bacterial pathogens that cause severe morbidity and mortality is needed. To date, there are no licensed vaccines for *Streptococcus agalactiae* (Group B Streptococcus, GBS), but phase 2 clinical trials of immunization of pregnant women show promising results with 57–97% infant seroresponses with immunoglobulin(Ig) G titres above the threshold associated with protection[10]. Despite the WHO recommendations on screening-guided intra-partum antibiotic prophylaxis, GBS claims an estimated 90,000 new-born deaths globally that could be reduced by 90% with vaccination[11,12]. However, there is little understanding regarding the cellular characterization that would mediate an effective vaccine-induced immune response due to the limitations of analytical tools.

The importance of long-lasting memory B cells (Bmem) in infection and vaccination is widely recognized as B cells are responsible for antibody-mediated immunity against microbial surface molecules, and this is highly effective in protecting against extracellular bacteria. Moreover, T cells are commonly not reactive to PS antigens as these are not presented on major histocompatibility complex molecules by antigen-presenting cells, and PS-specific memory is attributed to B cells. PS antigens can confer long-lasting humoral immunity and B cell memory in a T-cell-independent manner[13]. However, for a high-affinity Bmem response, B cells require assistance from T helper cells to activate affinity maturation via somatic hypermutation and clonal selection. By coupling PS to immunogenic carrier proteins, PS conjugate vaccines can elicit T cell help and improve Bmem responses, which explains their success over purified PS vaccines that lack protein-conjugation[14]. Despite the steep decline in numbers of circulating Bmem after the vaccination peak, PS-specific B cells can be detected, and boosted, years after vaccination with conjugate vaccines[15–17]. Studies with an experimental human pneumococcal carriage model show that serotype 6B-specific IgG[+] Bmem, but not circulating antibody levels, correlate with protection against Spn colonization[18]. The characterization of antigen-specific B cells by cytometry has provided valuable insight for therapy and vaccine development, and could be used to predict the longevity of vaccine responses[19].

Currently, PS-specific B cell kinetics elicited by bacterial infection and vaccination remain incompletely characterized due to analytical limitations related to the biochemical nature of PS (as reviewed previously[20,21]). The current gold standard to examine PS-specific Bmem is using enzyme-linked immunosorbent spot (ELISpot), in which Bmem are differentiated in vitro to plasmablasts followed by detection of secreted antigen-specific antibodies[21]. This method is inherently a semi-quantitative and indirect measurement of Bmem. It is also low-throughput and does not allow B cell subset assessment nor phenotypical characterization, which may withhold valuable information for evaluating protective immunity[21]. Despite the greater demand for expertise and costs, flow cytometry approaches are increasingly used and can overcome the limitations of ELISpot. The use of fluorescently labelled protein antigens has become increasingly sought-after for studying antigen-specific B cell immunity via cytometry[22]. However, this assessment has been problematic for carbohydrate antigens as they vary widely in structure and biochemistry and often lack free amines, which are commonly used to conjugate antigens to protein carriers. This makes a universal labelling approach difficult. Therefore, during the production of conjugate vaccines, PS are first treated with either reducing (sodium metaperiodate) or cyanylating (1-cyano-4-dimethylaminopyridinium (CDAP) tetrafluoroborate) reagents, after which conjugation to a carrier protein is performed[23]. Recently, various methods were published to study PS-specific Bmem via cytometric analysis[16,24–28]. However, these methods sometimes lack adaptability due to the dependency on the structure of the PS of interest and they have a limited high-throughput potential, as currently described protocols were only used for few serotypes of one bacterial species. Because of this, to date the heterogeneity in immune responses cannot be assessed with existing tools. Moreover, currently described assays for PS-specific B cells do not use combinatorial staining, in which antigens are labelled with multiple fluorochromes to increase throughput, reduce noise and improve assay sensitivity, as was shown in studies with antigen-specific T cells[29,30]. Recently, this approach was also applied to study B cells specific for six protein antigens[31].

Here, we aimed to develop a robust and flexible method for simultaneous quantification and in-depth characterization of PS-specific Bmem for multiple different serotypes from different pathogens. We optimized conditions for creating multimers of biotinylated PS and streptavidin (SA) that can be used for combinatorial staining and concurrent detailed phenotyping of PS-specific B cells using spectral flow cytometry. We validated the potential of this method by studying B cells specific for 14 different Spn PS serotypes from pneumococcal conjugate vaccine (PCV) 13-immunized and unvaccinated donors, as well as B cells specific for five different GBS PS serotypes from South African pregnant women and Dutch donors. This revealed differences in PS-specific B cell phenotypes associated with serotype-specificity, vaccination status and donor population.

## Results

**Validation of biotinylation and multimerization of polysaccharides.** To create fluorescently labelled PS, we first cyanylated purified capsular PS with CDAP (Fig. 1a)[32]. After cyanylation, we biotinylated the activated PS through coupling to amine-PEG$_3$-biotin, followed by purification using spin columns, as described[32]. We biotinylated PS of 14 Spn serotypes, the 13 covered in PCV13 (PS1, PS3, PS4, PS5, PS6A, PS6B, PS7F, PS9V, PS14, PS18C, PS19A, PS19F and PS23F) plus the non-vaccine type PS15B, and confirmed successful biotinylation for each included PS with a biotin enzyme-linked immunosorbent assay (ELISA) compared to non-biotinylated PS6B (Fig. 1b). To test whether the cyanylation and biotinylation steps compromised PS epitopes, we performed a competition ELISA. Using serum from a PCV13-vaccinated donor, we assessed whether serotype-specific antibodies could be similarly blocked by preincubation with biotinylated PS or non-modified PS (Fig. 1c). PS modification did not alter antigenicity for 10 out of 14 PS, and led to a decrease in antigenicity for PS1 (40%), PS3 (18%), PS19A (33%) and PS19F (11%), compared to the unmodified PS. Thus, cyanylation and biotinylation minimally altered the

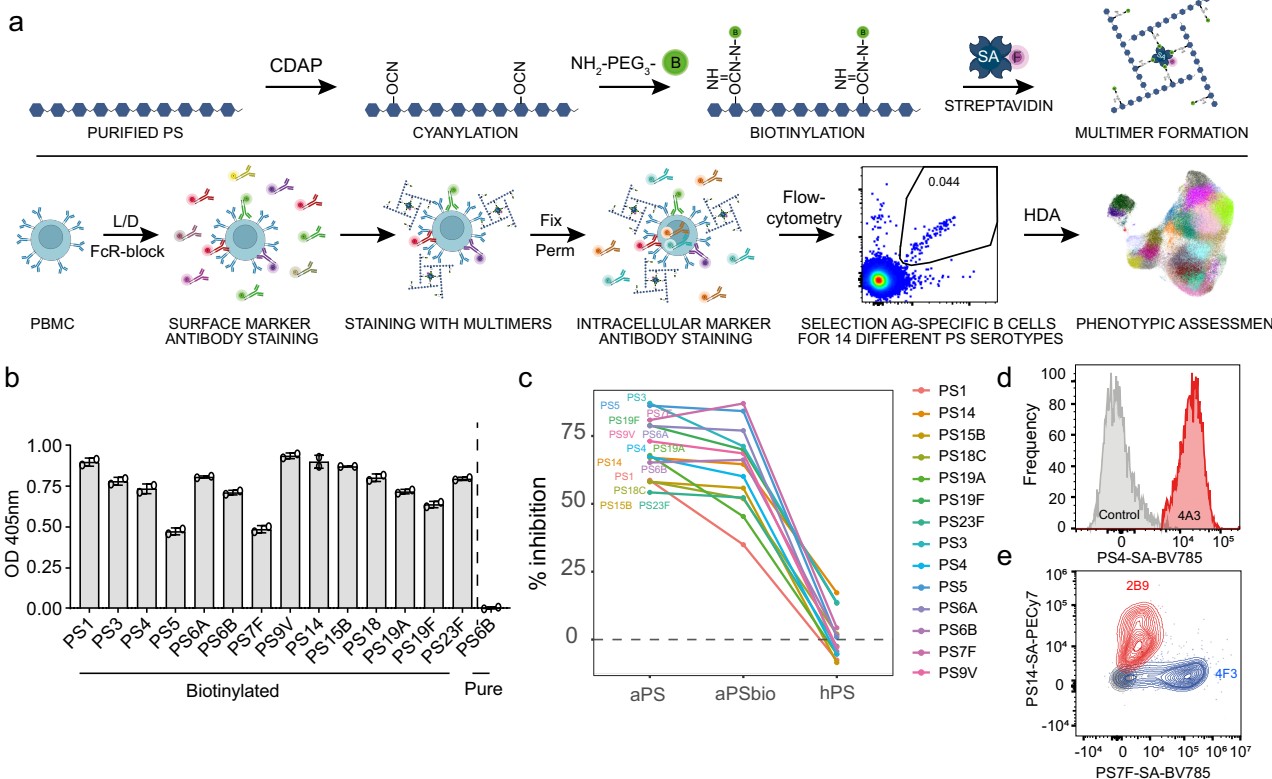

**Fig. 1 Validation of pneumococcal PS biotinylation and multimerization. a** Workflow, including the consecutive steps for the generation of fluorescently labelled PS-multimers (top) and the staining and detection of serotype-specific B cells (bottom). Acronyms used: PS (polysaccharide), CDAP (1-cyano-4-dimethylaminopyridinium tetrafluoroborate), NH₂-PEG₃-B (EZ-link amine polyethylene glycol biotin), SA (streptavidin), F (fluorochrome), PBMC (peripheral blood mononuclear cells), L/D (live/dead staining), FcR-block (FcReceptor blocking), fix (fixation), perm (permeabilization), Ag (antigen) and HDA (high-dimensional analysis). **b** Efficacy of PS biotin incorporation, as measured by biotin ELISA. For each PS, levels were measured in duplicate, indicated by symbols with average depicted by bars. Blank-corrected OD values are shown. Untreated PS6B (pure PS6B) is shown as negative control. **c** Competition ELISA showing the percentage of serotype-specific antibody blocking by biotinylated and unmodified PS. For each PS, inhibition of detection by preincubation of serum from a PCV13-vaccinated donor with autologous-unmodified PS (aPS), autologous-biotinylated PS (aPSbio) and heterologous-unmodified PS (hPS) is shown. Inhibition indicates the reduction in OD signal in preabsorbed versus non-preabsorbed serum. Average of duplicates is shown. **d** Histogram showing the fluorescent signal of serotype 4-specific clone 4A3 cells stained with PS4-SA(BV785) multimers (red) or unstained control (grey), normalized to mode. **e** Contour plot indicating the staining of serotype-specific clones 2B9 (PS14-specific, red) and 4F3 (PS7F-specific, blue) and 4F9 (negative control clone, grey) with both PS14-SA(PECy7) and PS7F-SA(BV785) multimers.

ability of serum PS-specific antibodies to bind for the majority of PS antigens.

We then generated multimers by combining the biotinylated PS with fluorescently labelled SA (Fig. 1a). Finally, peripheral blood mononuclear cells (PBMCs) were probed with the PS-SA multimers to detect antigen-specific binding, in combination with a 25-marker staining to identify and phenotypically characterize B cells via spectral flow cytometry. For quality control purposes, we verified that the signal intensity is distinguishable from the negative control by using PS-SA multimers and compensation beads coupled to Spn polyclonal serotype-specific antiserum. Multimer batches prepared on different days resulted in similar signal intensity and biotinylated PS remained stable after freeze-thawing (Supplementary Fig. 1a–c). The binding of the PS-SA multimers was specific, as beads labelled with serotype-specific antiserum did not recognize multimers of heterologous PS serotypes, e.g. antisera against PS6 did not bind PS9V multimers (Supplementary Fig. 1d). In all subsequent biotinylation and multimer preparations, we assessed biotin incorporation with biotin ELISA, antigenicity with competition ELISA and multimerization with PS-specific antiserum and compensation beads as quality control steps.

Next, we tested whether B cells could be stained with the PS-SA multimers, using serotype-specific immortalized human B cell clones obtained by selective expansion of BCL6/Bcl-xL-transduced peripheral blood PS-specific B cells[33]. PS4-SA multimers stained PS4-specific B cell line clones (clone 4A3), but not the control B cell line that lacked this specificity (Fig. 1d). To confirm the specificity of binding, we probed three different B cell clones that were specific for either PS7F (clone 4F3) or PS14 (clone 2B9), as well as a negative clone, with both PS7F-SA and PS14-SA multimers (Fig. 1e). The PS7F- and PS14-specific clones bound their corresponding serotype exclusively, meanwhile the control clone was negative for either of the multimers. These findings confirmed a serotype-specific recognition of PS without cross-reactivity. The PS-specific human B cell clones were also used to assess the optimal staining conditions, including duration, temperature, concentration and PS:SA ratio, and conditions were confirmed with PBMCs (Supplementary Fig. 1e–h). A PS:SA ratio of 4:1 for multimerization and 30 min staining on ice at a concentration of 5 µg/mL led to the highest quality signal with minimal background or interaction between co-stained PS.

**Validation of combinatorial staining and specificity of multimers.** To enhance the number of PS serotypes that can be simultaneously detected and to reduce non-specific binding, we applied combinatorial PS-SA staining (Fig. 2a). By preparing two multimer

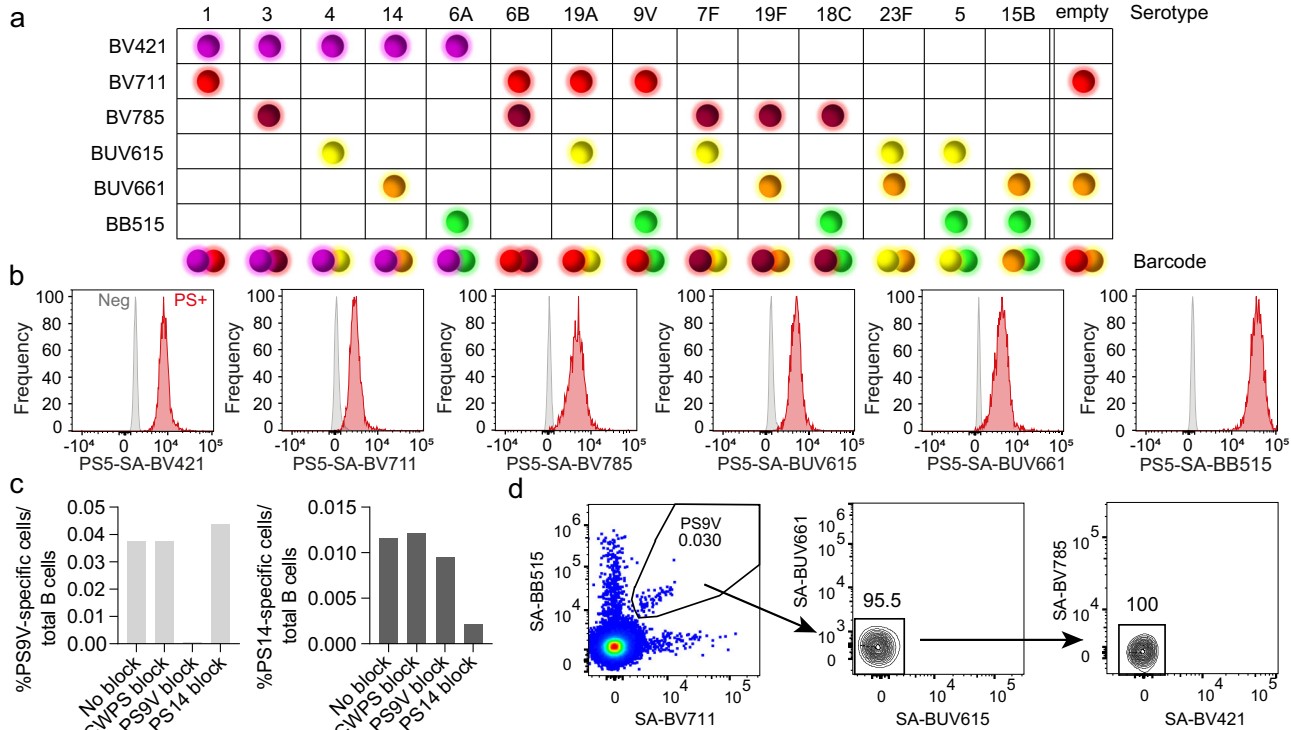

**Fig. 2 Combinatorial staining with PS-SA multimers leads to specific detection of PS-specific B cells. a** Barcode method of combinatorial staining patterns for PS-SA multimers. The six fluorochromes are indicated in rows and the various serotypes in columns, including all PCV13 serotypes, PS15B and an empty colour combination for background control. **b** Histograms showing fluorescent signal of compensation beads alone (Neg, grey) or coupled to PS-specific antisera and stained with of PS5-SA multimers using six different fluorochromes (PS+, red), normalized to mode. **c** Specificity of PS-SA multimers binding to serotype PS9V- (left panel) and PS14-specific cells (right panel). PBMCs from one donor (3 weeks after PCV13 vaccination) were pre-blocked for 30 min with 100 μg/mL cell wall polysaccharide (CWPS), unmodified PS9V or unmodified PS14, followed by PS-SA multimer staining. Pre-gated viable B cells (CD19+CD3−CD56−) were analysed for PS-specific double-positive cells in addition to the absence of signal positivity for other PS-SA multimers, to establish the frequency of PS-specific cells. **d** Non-blocked PBMCs from one donor (3 weeks after PCV13 vaccination) were gated for non-cross-reactive PS9V-specific cells (BV711+BB515+BUV615−BUV661−BV421−BV785−). The percentage of parent is indicated above the quadrants.

batches with different fluorochrome combinations for each PS, up to 15 different serotype-specific populations can be identified with 6 different fluorescent labels. We left one combination intentionally blank (empty) to assess the assay background. Combinatorial staining of B cell clones showed similar staining intensities compared to the use of a single fluorochrome; however, slightly lower frequencies (Supplementary Fig. 1i). The use of double staining may lose resolution for low BCR-expressing cells, as receptor availability is divided by half. Therefore, six bright fluorochromes (BB515, BV421, BV711, BV785, BUV615 and BUV661) with good positive-negative signal separation were selected out of PS5-SA multimers prepared with 10 different fluorochromes using beads coupled to serotype-specific antiserum (Fig. 2b and Supplementary Fig. 1j).

To demonstrate the binding specificity of the PS-SA multimers on PBMCs, we quantified the PS-specific cells after pre-blocking with purified PS from two different serotypes. PBMCs of a PCV13-vaccinated donor were incubated with either unmodified cell wall polysaccharide (CWPS, a common contaminant of PS[34]), PS9V or PS14 and subsequently probed with PS-SA multimers to assess the frequencies of PS9V- and PS14-specific B cells (Fig. 2c and Supplementary Fig. 1k). Preincubation with autologous PS reduced the frequency of detected PS-specific cells (from 0.038% to 0.0005% for PS9V; from 0.012% to 0.002% for PS14). Therefore, PS from both serotypes successfully pre-blocked PS-specific B cells prior to probing with the multimers. Of note, pre-blocking did not reduce the frequency of the heterologous PS, therefore confirming antigen-specific binding. In addition, pre-blocking with CWPS did not affect the frequency of cells, showing that identified cells are not

cross-reactive against the CWPS. A further indication of specific staining was that PS-SA multimers exclusively stained B cells, and not T cells, NK cells or monocytes (Supplementary Fig. 1l). To further improve the specificity of detection, we defined cells as PS-specific if cells were positive for exclusively two out of six fluorochromes, as barcoded per PS (Fig. 2a, d). Cells binding additional multimers could be indicative of, for example, cross-reactivity, non-specific binding of PS to a pattern-recognition receptor or specific binding to a contaminant of the PS or to the SA.

Our next goal was to verify the staining concentration for PBMCs. We observed similar PS-specific cell frequencies after further increasing the staining concentration of the PS-SA multimers to 9 μg/mL, confirming 5 μg/mL as the optimal staining condition (Supplementary Fig. 2a, b). We also compared whether different fluorochromes would impact the frequency of identified cells, which would prevent the direct comparison of frequencies across serotypes. To address this, we stained PBMCs from three donors with unknown vaccination status with PS7F-, PS19F- and PS15B-SA-multimers that contained non-overlapping colour combinations: SA-BV421/BUV615, SA-BV711/BUV661 and SA-BV785/BB515 (Supplementary Fig. 2c, d). The detected frequencies were independent of the colour combination used, indicating that fluorochrome choice did not affect the experimental outcome.

**Detection of naturally acquired and vaccine-elicited PS-specific B cells in PBMCs.** One of the main advantages of flow-based assays is the opportunity for comprehensive phenotypic characterization of

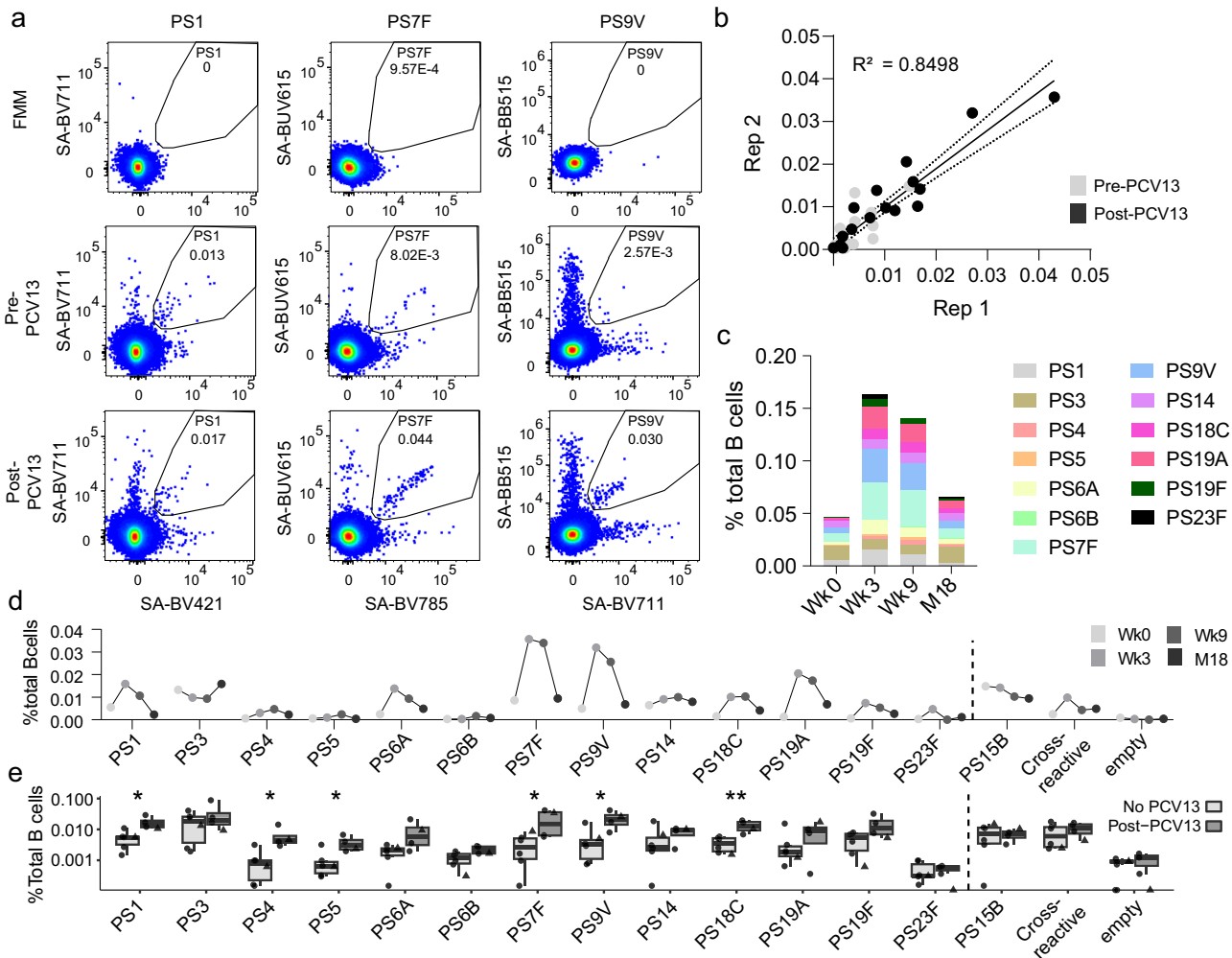

**Fig. 3 PCV13 vaccination expands the frequency of PS-specific B cells. a** Gating and frequencies of PS1, PS7F and PS9V-specific B cells from PBMC at baseline and 3 weeks post-PCV13 vaccination. Fluorescence minus multimer (FMM) control sample contains cells from all timepoints combined and stained with the full staining, with the exception of the multimers. Cells were pre-gated for viable B cells (CD19+CD3-CD56-). **b** Correlation between two independent experiments (Rep 1 and Rep 2). The black lines indicate linear regression ($R^2 = 0.849$) and the 95% confidence interval. Statistical analysis by Pearson correlation. PBMC samples from before and 3 weeks after PCV were included. **c** Frequency of accumulated 13 PS-specific cells, from one donor at baseline (Wk0) and 3 weeks (Wk3), 9 weeks (Wk9) and 18 months (M18) after PCV13 vaccination. **d** Longitudinal analysis showing the PS-specific B cell frequencies from (**c**) over time (represented in various shades of grey). **e** Frequency of PS-specific B cells in PBMCs from 5 donors with no known PCV13 vaccination (no PCV13, light grey, circles) and 3 donors around 16 weeks after PCV13 vaccination (post-PCV13, dark grey, circles), in combination with samples from or longitudinal donor at baseline (no PCV13, light grey, triangles) and 9 weeks post-PCV13 (post-PCV13, dark grey, triangles). Statistical analysis by Mann–Whitney test (*$p < 0.05$, **$p < 0.01$).

antigen-specific cells. We established a 25-marker antibody panel in conjunction with the PS-SA multimers for an in-depth characterization of serotype-specific B cells (Supplementary Table 1)[35]. We applied our assay to study the effect of PCV13 vaccination on the frequency and phenotype of Spn serotype-specific Bmem response. Therefore, we collected PBMCs from a donor at baseline and at 3 timepoints after PCV13 vaccination (3 weeks, 9 weeks and 18 months). At week 3 after vaccination, B cells against 12 out of 13 PCV13-included serotypes showed increased frequency (Fig. 3a–d). As expected, there was no increase of PS15B-specific B cells, as this serotype is not covered by PCV13. There was a large variation among induced frequencies of PS-specific B cells, with strong increases for PS19A (median 1060%), PS9V (460%), PS19F (360%) and PS7F (330%) at 3 weeks after PCV13, while others such as PS14 (40%) and PS3 showed poor or no expansion. Notably, for this donor, PS3-specific pre-vaccination levels were the highest out of all serotypes, possibly as colonization by this serotype is prevalent among adults in The Netherlands[36].

A median of 88% (83.5–97.2%) of multimer-positive cells were considered serotype-specific, being exclusively positive for two out of six fluorochrome channels, with cross-reactive cells defined as a separate population (Supplementary Fig. 2e). Technical duplicates showed that the frequencies were alike among different measurements ($R^2 = 0.961$) (Supplementary Fig. 2f). To examine the reproducibility of our experiments, we stained the same samples three times across independent experiments with different batches of multimers, showing highly concordant measured frequencies ($R^2 = 0.849$ for batch 1 and 2 shown as example) (Fig. 3b).

We performed a longitudinal analysis to examine the post-vaccination kinetics of PS-specific B cells in this donor (Fig. 3c, d). Compared to the pre-vaccination baseline, overall accumulated PCV13 PS-specific B cell frequencies increased 3 weeks after vaccination by approximately 250% for this donor. After peak expansion, total PS-specific cell numbers started waning, remaining 200% higher than baseline at 9 weeks, and 40% at 18 months.

To study PCV13-induced B cells in more individuals, we conducted a cross-sectional analysis including blood samples derived from five blood bank donors (vaccination status unknown, but likely negative given the donor population that consists of healthy adults) and three additional PCV13-vaccinated donors (collected around 16 weeks after vaccination) that we combined with our longitudinal donor before and 9 weeks after vaccination (Fig. 3e). While this validation analysis was cross-sectional and we missed the expansion peak at the timepoint of the vaccinated donors, an approximate median 145% increased PS-specific B cell frequencies were apparent in vaccinated individuals. Notably, as seen in our longitudinal donor, PS7F (1100%; $p = 0.04$) and PS9V (630%; $p = 0.02$) serotype-specific B cells again showed the largest differences in frequency, while PS3 frequencies (no difference of median frequency, $p = 0.76$) were similar between vaccinated and unvaccinated individuals. The negative control of all analysed samples was less than 0.0013%, indicating the background noise and defining the limit of sensitivity of the assay at around 1 per 100,000 B cells (Supplementary Fig. 2g). In non-vaccinated individuals, frequencies of PS4, PS5 and PS23F-specific cells were around this level. In vaccinated individuals, PS4- and PS5-specific B cells became significantly more abundant, while frequencies of PS23F-specific cells remained close to the limit of detection.

**High-dimensional analysis and phenotypic characterization of PS-specific cells.** To phenotypically characterize the total of 5,935 identified serotype PS-specific B cells across all donors from our cross-sectional and longitudinal analyses combined, we applied Uniform Manifold Approximation and Projection (UMAP) dimensionality reduction analysis, based on 23 included B cell markers (Fig. 4a and Supplementary Fig. 3). PS-specific B cells were most frequent in the IgG$^+$ compartment, while notably 50–70% of PS1- and PS3-specific cells were IgD and IgM double positive (Fig. 4a–c and Supplementary Fig. 3b).

Subsequent clustering of the B cells with FlowSOM identified a total of 35 elbow clusters (Fig. 4b). The most abundant clusters for PS-specific B cells were elbow 3 and 4, which are both class-switched memory cells (IgG$^+$CD27$^+$CD45RB$_{MEM55}$$^+$) and differ by the expression of the activation marker and death receptor CD95 (Fas) in cluster 4 (Fig. 4d). PS1- and PS3-specific cells, two serotypes that are associated with persistent invasive disease and poor vaccine response[7,37], were comparatively less abundant in clusters 3 and 4.

To statistically assess whether serotypes or vaccination history were associated with PS-specific B cell phenotypes, we performed bootstrap on a regression model including serotypes and vaccination status with B cell clusters. PS1- and PS3-specific cells were significantly associated with clusters 2 (PS3), 22 (PS3) and 24 (PS1 and PS3) after correcting for multiple testing, showing an enriched IgM$^+$CD45RB$^+$ memory phenotype (Fig. 4d and Supplementary Data 2). Serotype-specificity was therefore associated with distinct B cell phenotypes. Looking at the overall effect of vaccination, clusters 3 (IgG$^+$CD27$^+$CD45RB$^+$CD95$^-$) and 7 (IgA$^+$CD27$^+$CD45RB$^+$CD73$^+$CD38$^{low}$), among others, were significantly enriched after vaccination (Fig. 4e and Supplementary Table 2). Frequencies for most serotype-specific cells were increased within these clusters (Fig. 4f). Although heterogeneity between serotypes was clear, these analyses did not take individual PS-specific effects by vaccination into account via interaction terms, as we were not powered to do so with our limited sample size.

**Application of PS-specific multimers for a heterologous bacterial species.** We verified the adaptability of our assay by analysing serotype-specific B cells against GBS, a different pathogen for which PS-specific B cells are of interest. We employed the

same methodology presented for Spn to biotinylate 5 serotypes of GBS (PSIa, PSIb, PSII, PSIII, PSV) (Supplementary Fig. 4a). The cyanylation and biotinylation of GBS PS showed minimal effect on antigenicity when assessed with competition ELISA (Supplementary Fig. 4b). Interestingly, our ELISA results revealed cross-reactivity between PSIb IgG antibodies with PSIa and PSV, and PSIII IgG antibodies with PSIa (Supplementary Fig. 4b, c). This was potentially caused by non-specific binding to contaminants in the purified GBS PS preparations. However, antiserum-covered compensation beads exclusively bound autologous PS-SA multimers and were unable to bind heterologous PS-SA multimers (Supplementary Fig. 4d, e). Moreover, PBMCs stained with only PS-SA multimers for serotypes PSIb and PSIII, which suffered most from cross-reactivity in the competition ELISA, resulted in similar frequencies compared to staining for all GBS PS serotypes combined (Supplementary Fig. 4f). Combined, these findings suggest that the observed cross-reactivity in ELISA does not affect the binding of B cells to non-specific PS-SA multimers.

We used combinatorial staining with four different fluorochromes to stain for cells specific against the five GBS serotypes, including again an empty combination to assess assay background and lower limit of detection (Supplementary Fig. 4g). GBS PS are nearly 10-fold smaller in size (60–130 Kda) comparatively to Spn PS (600–1200 Kda)[38,39], and titration indicated a lower concentration of 2.5 μg/mL is required for staining with optimal signal detection and minimal background (Supplementary Fig. 4h). In contrast to Spn PS-SA multimer staining, we observed substantial frequencies of cross-reactive B cells recognizing more than one GBS serotype (defined as a separate population), and monocytes were able to bind the GBS PS multimers (Supplementary Fig. 5a, b). GBS PS contain side chains terminated with sialic acid in α2-3 linkage and immune cells, including B cells, contain multiple receptors for sialic acids, such as sialic acid-binding immunoglobulin-type of lectins (Siglec)-5, Siglec-6 and Siglec-9[40]. Pre-blocking Siglec receptors with recombinant antibodies indeed reduced the frequency of cross-reactive B cells ($p = 0.023$), with Siglec-9 being the most effective, in line with its specificity for α2-3-linked sialic acids (Supplementary Fig. 5b, c). Therefore, we proceeded with adding a pre-blocking step of Siglec-9 to our PBMC staining protocol to reduce cross-reactivity. With PSIa and PSV pre-blocking experiments, we validated the specificity of B cells for multimer binding (Supplementary Fig. 5d). Technical duplicates showed comparable PS-specific B cell frequency readouts ($R^2 = 0.906$), indicating again minimal intra-assay variation (Supplementary Fig. 5e, f). Comparable to the Spn readings, we observed a background frequency of less than 0.001% (Supplementary Fig. 5g).

**Detection and characterization of naturally acquired GBS PS-specific B cells in samples from South African and Dutch donors.** Assessing geographical differences in immune responses is imperative for vaccine implementation, especially given the variation in disease incidence and immunization efficacy[41]. We therefore stained PBMCs from 8 anonymous Dutch donors and 22 South African pregnant women donors (representing areas with low versus high incidence of GBS invasive disease, respectively). The frequency of serotype-specific B cells for PSIa, PSII and PSIII was significantly higher in South African donors compared to Dutch, while PSIb- and PSV-specific B cells showed similar frequencies between both populations (Fig. 5a).

We then proceeded to characterize the PS-specific cells via differential expression of 24 cellular markers (Supplementary Table 3 and Supplementary Fig. 6). The majority of GBS PS-specific B cells were IgG$^+$ or IgM$^+$ memory cells (CD27$^+$CD45RB$^+$) (Fig. 5b–d and Supplementary Fig. 5h, i). FlowSOM clustering resulted in a total of 33 phenotypically distinct elbow clusters

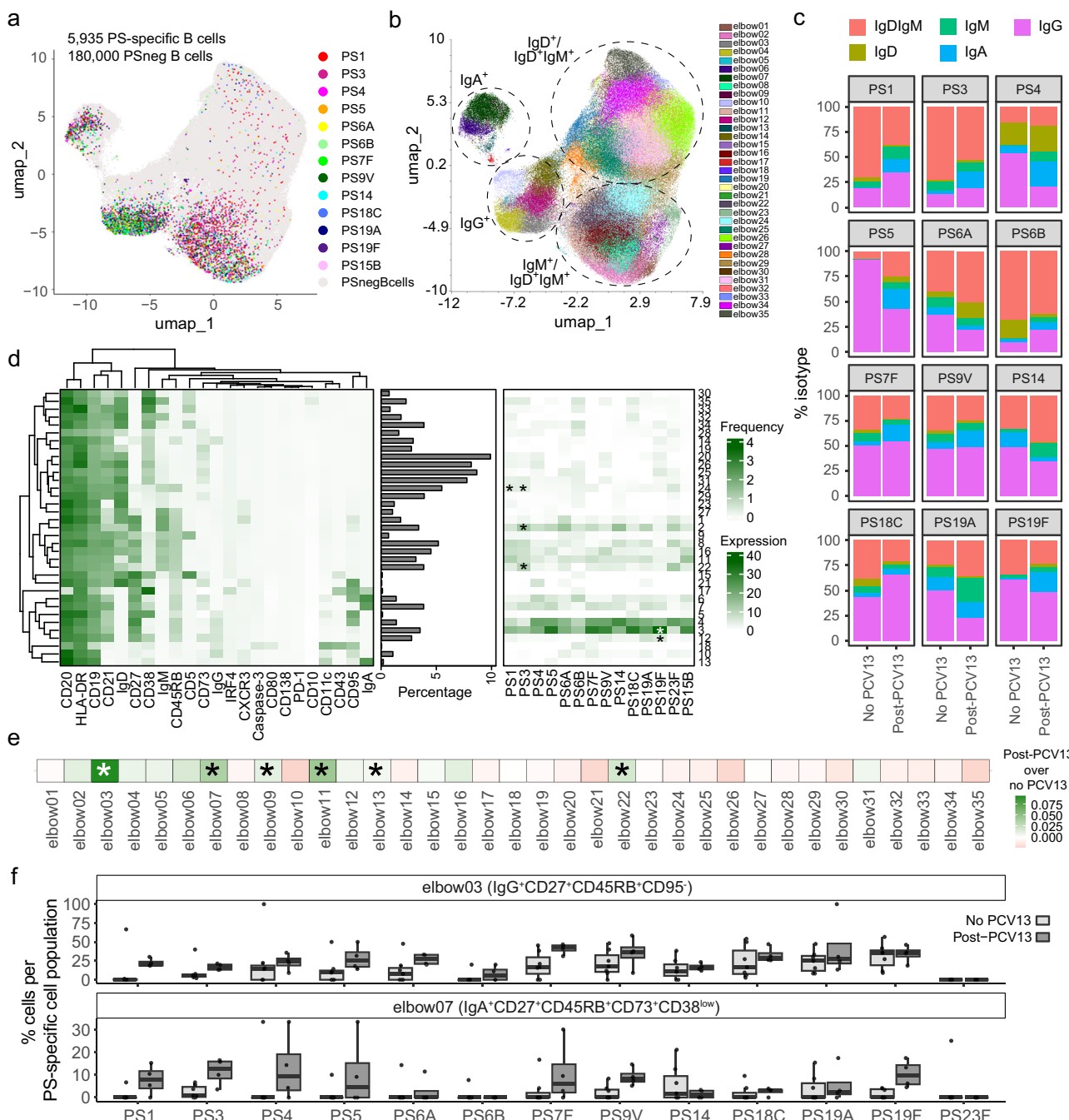

**Fig. 4 Phenotyping of PS-specific cells. a** Distribution of 5935 PS-specific B cells against 14 serotypes (coloured) and 180,000 PS-negative cells (PSnegBcells; light grey) overlaid on UMAP, using cells from all donors and including experimental repeats from our longitudinal and cross-sectional analysis. In total, 18 samples are shown. **b** UMAP projecting the 35 FlowSOM-clusters from all cells in (**a**). **c** Frequency of each IgD+IgM+, IgD+, IgM+, IgG+ and IgA+ PS-specific B cells from 5 assumed non-vaccinated donors (no PCV13), one donor at baseline (no PCV13) and 4 PCV13-vaccinated donors. Donors with <5 cells per PS-specific population were excluded. **d** Heatmaps indicating the median expression of 23 surface markers that characterize each B cell cluster (left), the frequency of each cluster in total PS-negative B cells (middle) and the distribution of PS-specific cells among the clusters (right, filtered for ≥5 cells per PS-specific population per donor). Stars indicate which serotypes are significantly associated with a cluster, assessed in all samples from (**b**), using an upper-tailed test based on a linear regression model with wild bootstrap simulation at 9999 resamples. Bonferroni correction for multiple testing was employed. **e** Heatmap showing the association between PCV13 vaccination and cluster frequency, assessed with an upper-tailed test based on a linear regression model with wild bootstrap simulation at 9999 resamples and Bonferroni correction. **f** Frequencies of clusters 3 and 7 per PS-specific cell population, from 5 donors assumed with no PCV13 (no PCV13; light grey), one donor pre-PCV13 (no PCV13; light grey) and 4 PCV13-vaccinated donors (post-PCV13; dark grey).

(Fig. 5c, e and Supplementary Fig. 5j). Compared to Spn, we observed more phenotypic diversity among GBS PS-specific cells. Cluster 21 (IgG+CD27+CD45RB+) and clusters 10 and 20 (both IgD+IgM+CD27+CD45RB+) were most abundantly represented

among the PS-specific cells, except for PSIa cells. These were minimally present in clusters 10 and 20, and often found among IgGlowCD27+CD95+CD73+ (clusters 24 and 27) and IgD+IgM+ LAIR1+CD5+ (cluster 2) Bmem.

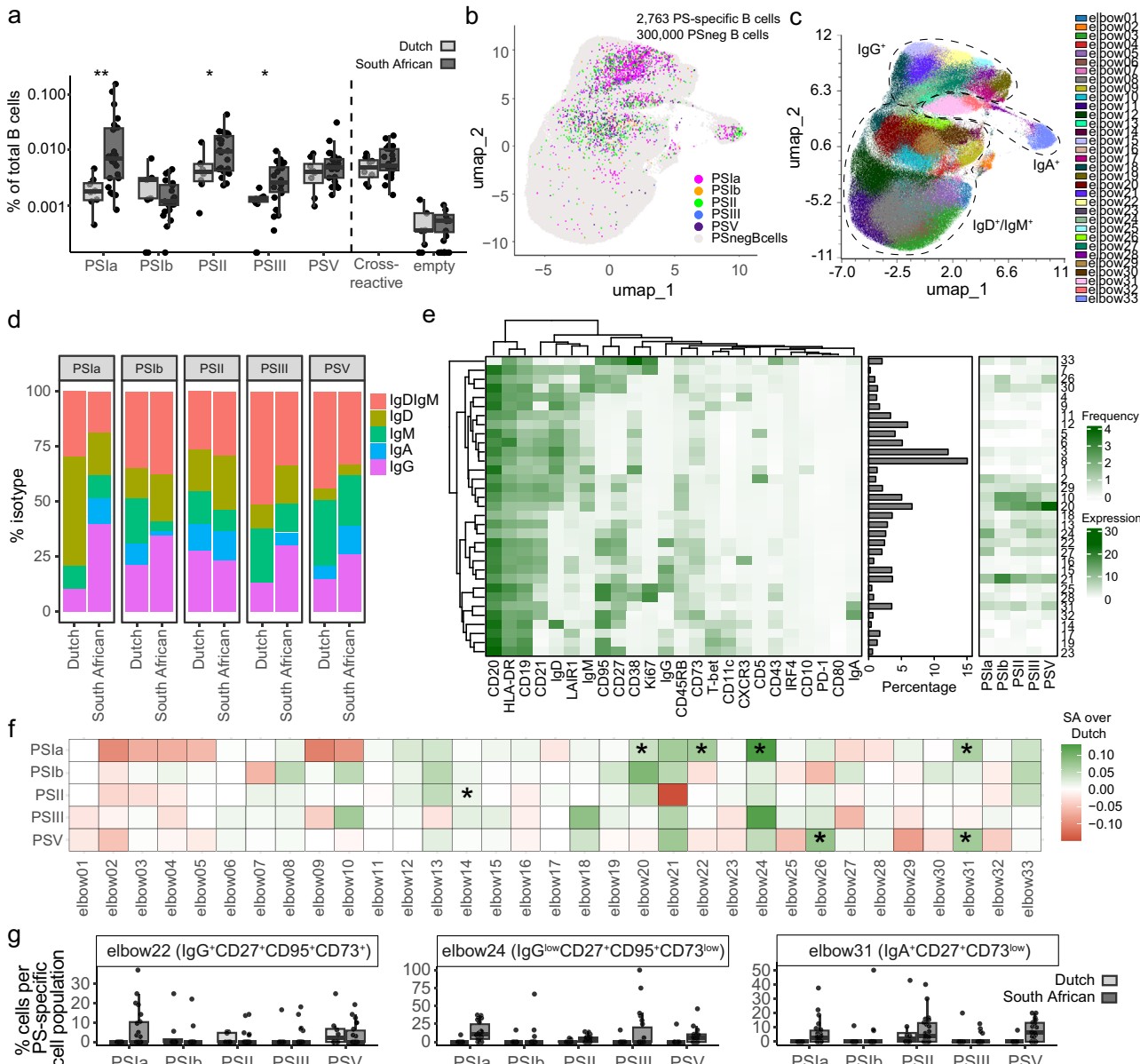

**Fig. 5 GBS serotype-specific cells. a** Frequencies of serotype-specific B cells in PBMCs from 8 anonymous Dutch (light grey) and 22 South African pregnant women donors (dark grey). Statistical analysis was performed with the Mann–Whitney test. **b** Distribution of 2763 PS-specific B cells against 5 serotypes (coloured) and 300,000 PS-negative cells per donor (PSnegBcells; light grey) overlaid on UMAP. In total, cells from 30 donors were used. **c** UMAP projecting the 33 FlowSOM-generated elbow clusters. **d** Frequency of each IgD+IgM+, IgD+, IgM+, IgG+ and IgA+ PS-specific B cells in PBMCs from 8 Dutch and 22 South African donors. Donors with <5 cells per PS-specific population were excluded. **e** Heatmaps with a median cellular expression of 24 markers (left) on each cluster from (**c**), the frequency of each cluster among PS-negative cells (middle) and the distribution of PS-specific cells among the 33 clusters (right, filtered for ≥5 cells per PS-specific population per donor). **f** Heatmap showing the difference between serotype-specific cells associated with each cluster between South African (SA) and Dutch donors, assessed with a two-tailed test based on a linear regression model with wild bootstrap simulation at 9999 resamples. Bonferroni correction for multiple testing was employed. Stars indicate a significantly increased frequency of cluster-associated PS-specific B cells in South African donors (estimate >0.01). **g** Frequencies of clusters 22, 24 and 31 per PS-specific cell population in Dutch (light grey) and South African donors (dark grey) (*$p < 0.05$, **$p < 0.01$).

To compare PS-specific B cell phenotypes between Dutch and South African donors, we applied regression model analysis with bootstrap, looking at the interaction between PS and donor population. South African donors showed increased IgG class-switched PSIa-specific B cells compared to Dutch donors (Supplementary Fig. 5k and Supplementary Table 4). At a cluster level, for PSIa, higher frequencies of clusters 20, 22 (IgG+CD27+CD95+CD73+), 24 (IgGlowCD27+CD95+CD73low) and 31 (IgA+CD27+CD73low) were present among South African

donors, compared to Dutch (Fig. 5f, g and Supplementary Table 5). PSV, despite having similar overall frequencies between sites, showed increased frequencies in clusters 26 and 31 in South African donors.

## Discussion
Here, we developed and validated a robust and flexible method to quantify and phenotypically characterize serotype-specific B cells

heterogeneity using multimers of PS antigens. Our approach may offer great utility for PS-vaccine optimization studies. With a relatively small number of donors and low sample volume, optimal conjugation strategies, dosing regimens and novel formulations could be tested using this here-described tool. This is of particular use for the paediatric population, where limited amounts of blood can be collected. Moreover, due to the high-throughput potential of this method, it may be of great interest for the validation of novel multi-valent vaccines.

To validate and standardize the procedures of our method, we implemented multiple quality control steps that were performed with each antigen batch preparation. First, to validate the successful incorporation of biotin into the activated PS, we performed a biotin ELISA. Then, we tested whether the biochemical alteration of the PS altered the antigenicity with the competition ELISA. Lastly, we validate effective PS-SA multimerization with compensation beads coupled to PS-specific antiserum via spectral flow cytometry. We included technical duplicates and performed multiple experimental repeats in our analysis to assess the reproducibility of our experiments. We observed high concordance among technical duplicates, and between independent experiments performed with the same or different multimer preparations. We therefore show that the procedures and materials are robust and allow for representative results among experiments. The described quality control steps are of use to other groups establishing this assay at their own facility, or for adapting the assay to other PS antigens. Indeed, this approach can easily be adapted to cover more PS antigens, or those from different bacterial sources including non-capsular PS antigens, as long as purified antigen is available.

Epidemiological studies have demonstrated that PCV-elicited immunity varies among covered PS serotypes and appear less effective for serotype 3[7–9,42]. With our assay, we showed that Spn PS1- and PS3-specific cells at baseline were abundant IgD+IgM+ non-switched memory cells. These findings suggest that serotypes 1 and 3 are ineffective in generating protective IgG+ Bmem during colonization. Spn serotype 3 bacteria have the unique ability to shed the capsular PS as a mean to evade the hosts' immune response[43], which could result in poor stimulation and class-switching of PS3-specific B cells. This is in agreement with cross-sectional serological studies from Malawi, following PCV introduction[44]. However, we were surprised to also detect few isotype class-switched PS1-specific B cells, as unlike other serotypes, PS1 is a zwitterion reported able to stimulate T helper cell responses and IgG production in mice[45]. This may be explained by the independent generation of antibody-producing plasma cells and Bmem, as IgG class-switched B cells favour the differentiation of plasma cells instead of memory responses[46]. However, additional serological analysis would be required for confirmation, highlighting the added value of studying Bmem in complement to antibodies. In our analysis, one dose of PCV13 in adults was related to higher levels of class-switching to IgA and IgG. This corresponds with studies showing increased mucosal IgA levels following repeated PCV13 vaccination in children, as these class-switched cells would then subsequently be boosted[47]. Comparing B cell phenotypes between pathogens showed that there were fewer class-switched GBS PS-specific Bmem compared to Spn PS-specific Bmem. A possible reason for this may be the presence of the terminal sialic acid residue on GBS PS. Mammals use sialylation to recognize self-antigens, a mechanism that is mimicked and exploited by various pathogens as an immune evasion strategy[48]. Siglecs on B cells act as inhibitory molecules that limit signalling after BCR stimulation and are crucial for regulating auto-immunity[49]. Moreover, sialic acids may also indirectly interfere with B cell class-switching by limiting T cell help, as Siglec-7 and Siglec-9 are negative regulators of T cell receptor signalling[50]. Alternatively, variation in B cell phenotypes

and class-switching could be related to the distinct anatomical niche of colonization (genital versus upper respiratory mucosa).

A limitation of this study is the sample size available for analysis. In particular, more longitudinal samples are required to understand possible serotype-specific effects of vaccination, and we therefore solely focused on the general effects of vaccination. Future studies should use longitudinal samples with colonization, and conjugated and unconjugated vaccination to allow the comparison of different immunization strategies. Also, only blood bank donor samples were available, meaning that vaccination status is unknown, although it is likely negative given the population that donates blood in the Netherlands. Finally, the South African donors were all pregnant women, while Dutch donors were anonymous and of unknown sex and age and it cannot be excluded that these might have an effect on the PS-specific B cell phenotype and frequency.

In summary, we described a robust method that allows the in-depth simultaneous characterization of antigen-specific B cells for up to 14 Spn or 5 GBS serotypes. We provided extensive validation, and report quality control steps that researchers can implement when applying this tool to these or other pathogens. Finally, we showed there is diversity within the PS-specific populations that may reflect cellular characteristics that underlie variation in PS-associated vaccine efficacy.

## Methods

**Inclusion and ethics**. All ethical regulations relevant to human research participants were followed. Blood from healthy donors with unknown vaccination status (likely non-PCV13) was collected with concentrated peripheral blood (Buffy coat) that was donated anonymously by healthy adult volunteers at Sanquin Bloodbank (Amsterdam, The Netherlands). PBMCs from PCV13-vaccinated adults were collected from the tissue bank at the Liverpool School of Tropical Medicine (HTA license: 12548). The GBS study with South African donors was approved by the Human Research Ethics Committee (Medical) of the University of the Witwatersrand (M090937). Informed written consent was obtained from all participating donors.

**PS biotinylation**. All GBS PS were kindly provided by Biovac South Africa and were purified by using tangential flow filtration method. All five PS contained no significant burden of residual protein and nucleic acid (all less than 3%) and molecular weight ranged from 109 to 196 kDa[51]. Biotinylation was performed as described previously[32]. Purified serotype-specific capsular Spn PS (SSI Diagnostica) and GBS PS (Biovac) were diluted to 1 mg/mL in milliQ $H_2O$ and cyanylated with CDAP (final concentration 1 mg/mL in acetonitrile) while vortexing for 30 s. Next, 0.2 M triethylamine (TAE; Sigma, T0886-100ML) in milliQ $H_2O$ was added to increase the pH of the solution, using final molarity of 2 mM TAE for neutral PS or 4 mM for acidic PS, while vortexing for 2 min. Finally, 20 mg/mL Pierce EZ-Link Amine-PEG₃-Biotin (Fisher, 11881215) in milliQ $H_2O$ was added to obtain a 1:1 ratio to the PS, and vortexed for another 10 s. The mixture was incubated for 3–4 h at room temperature to allow biotin incorporation. Next, PS were washed three times with milliQ $H_2O$ over a 100 kDa (Spn PS) or 30 kDa (GBS PS) Amicon spin filter (Merck, UFC510024 and UFC503024, respectively) centrifuging for 5 min at 12,000×g to remove free biotin. The filtered content was collected by equalling the final volume to the start volume with milliQ $H_2O$ and centrifuging the inverted filter in a collection tube for 2 min at 1000×g. Biotinylated PS were kept at 4 °C for use within 3 months, or at −80 °C for long-term storage.

**PS-specific multimers**. Biotinylated PS were diluted in PBS to obtain 400 pmol/mL (Spn PS) or 1600 pmol/mL (GBS PS) and kept

on ice in the dark until the addition of SA. Then, fluorochrome-conjugated SA was added gradually by mixing well in 10 separate steps with 10-min intervals in between, obtaining a final PS:SA molecular ratio of 4:1. Multimerization was performed in 1.5 mL Eppendorf tubes or in V-bottom 96-well plates, on ice in the dark and under sterile conditions. After SA addition steps number 5 and 10, a pulse spin was performed to pull down the content in the tubes or wells. Finally, D-biotin (Sigma; B4501-100MG) was added with a final molarity of 12.5 μM (Spn) or 28 μM (GBS) to ensure saturation of SA binding sites, and incubated for 30 min at 4 °C. Multimers were stored at 4 °C for use within 2 weeks and spun at 2500 × g for 5 min at 4 °C prior to use.

**Biotin ELISA.** Nunc Maxisorp 96-well plates were coated overnight with 25 μg/mL PS (for serotypes 1, 19 and 23 F) or 5 μg/mL PS (for all other serotypes). Plates were washed three times with 0.05% Tween20 in PBS (wash buffer for all washing steps). Next, the plate was blocked with 1% BSA for at least 3 h at room temperature. After blocking, plates were washed three times and 0.1 ng/mL SA-alkaline phosphatase (Thermo Fisher; 21324) in 0.1% BSA in PBS was incubated for 45 min at room temperature. Finally, the plates were washed three times and the reaction was developed with 100 μL p-Nitrophenyl phosphate (PNPP, Sigma; N-9389, reconstituted in 5 mL MilliQ per tablet) for 5 min and the results were read with a spectrometer plate reader at 405 nm.

**Competition ELISA.** The conditions for the competition ELISA were based on the WHO established guidelines[52–54]. For Spn, serum from a donor 3 weeks after PCV13 vaccination was diluted 1:100 in 0.5% BSA in PBS with 10 μg/mL cell wall PS multi (CWPS-multi; SSI Diagnostica; 68866; a frequent contaminant of PS-preparations). For GBS, serum from South African donors with confirmed presence of serotype-specific antibodies was pooled (final concentration 2–10 μg/mL per serotype-specific antibody) and diluted 1:100 in 0.5% BSA in PBS. The diluted sera was incubated overnight at 4 °C with nothing or 10 μg/mL of autologous-unmodified, heterologous-unmodified or autologous-biotinylated PS. In parallel, Nunc Maxisorp 96-well plates were coated overnight at 4 °C with autologous-unmodified PS at a final concentration of 25 μg/mL (for Spn serotypes 1, 19 and 23F), 5 μg/mL (for all other Spn serotypes) or 10 μg/mL (all GBS serotypes). The next day, the coated plate washed three times with 0.05% Tween in PBS (wash buffer for all washing steps), and thereafter blocked with 1% BSA in PBS for 3 h at room temperature. Then, the preabsorbed serum samples were added to the appropriate wells and incubated for 2 h at room temperature. After another three washes, 0.4 μg/mL polyclonal anti-human IgG (Sigma, A9544) detection antibody was incubated for 1.5 h at room temperature. Finally, three more washes were performed and the samples were developed with PNPP (Sigma, N-9389) in PBS for 15 min and the results were read with a spectrometer plate reader at 405 nm.

**Beads staining.** One drop of AbC total antibody compensation beads (Thermo Fisher; A10513) was incubated with 40 μL of PS serotype-specific rabbit antisera (SSI Diagnostica) for 15 min at room temperature. The beads were then washed twice with FACS buffer (2 mM EDTA and 0.5% BSA in PBS) and spun for 5 min at 600×g. The antiserum-coupled beads were then incubated with 11.5 μg/mL (Spn) or 3.5 μg/mL (GBS) PS-SA multimers in FACS buffer for 15 min at room temperature (Spn) or on ice (GBS). Multimers were centrifuged prior to use at 2500 × g for 5 min at 4 °C to pull down aggregates that may cause artefacts in the staining. The beads were then washed two times with 1 mL FACS buffer, spun for 5 min at 600×g, resuspended in 300 μL FACS buffer and acquired with a spectral flow cytometer (Cytek) on the same day.

**B cell clones.** Serotype-specific immortalized human B cell clones were previously obtained by selective expansion of BCL6/Bcl-xL-transduced peripheral blood Bmem, and specificity was verified via ELISA of culture supernatant (clones provided by AIMM Therapeutics)[33].

**Cell staining.** All antibodies, clones and dilutions used for staining are listed in Supplementary Tables 1 and 3. Five to ten million cryopreserved PBMCs in 10% dimethyl sulfoxide, and 20% foetal calf serum (FCS) in Roswell Park Memorial Institute (RPMI) 1640 were thawed by dropwise addition of 5 mL RPMI with 20% FCS prewarmed to 37 °C, and washed once more with the same medium. For Spn, the centrifugation steps after each wash were performed for 5 min at 4 °C and 450×g (before fixation) and 600×g (after fixation). Because we had lower cell numbers in the South African samples, the centrifugation for fixed cells was increased to 800×g for 7 min at 4 °C to reduce sample loss. For GBS samples, red blood cells were removed by incubating the PBMCs with 5 mL RBC lysis buffer (Apotheek AZL, 97930725) for 5 min at room temperature, and washed first with RPMI and then FACS buffer (2 mM EDTA and 0.5% BSA in PBS). FACS buffer was used for all subsequent wash steps, unless indicated otherwise.

Cells were stained for viability (Supplementary Tables 1 and 3) and blocked for CD16/32 with anti-human FC Receptor binding inhibitor (Thermo Fisher Scientific, 14-9161-73) in PBS for 15 min at room temperature (Spn) or 20 min on ice (GBS). Then, after one wash, the cells were first stained for extracellular surface markers with BD Brilliant stain buffer plus (BD, 566385) in FACS buffer for 15 min at room temperature (Spn) or 20 min on ice (GBS) and washed. For the GBS staining, cells were then blocked with 10 μg/mL anti-Siglec-9 (RnD systems, MAB1139-100) in FACS buffer to reduce unspecific binding of sialic acids on the PS. A sample fraction from all donors was combined up to 400,000 cells total for an FMM (fluorescence minus multimer) control that was stained for all markers except PS-SA multimers. The remaining cells were stained with 5 μg/mL (Spn) or 2.5 μg/mL (GBS) PS-SA multimers with BD stain plus in FACS buffer for 30 min on ice and washed. Multimers were centrifuged prior to use at 2500×g for 5 min at 4 °C to pull down aggregates that may cause artefacts in the staining. Then, a fixation/permeabilization (eBioscience, 00-5523-00) step was performed for 30 min at room temperature. Finally, cells were washed twice in permeabilization buffer (eBioscience, 00-5523-00) and stained for intracellular markers with 1X BD stain plus in permeabilization buffer for 30 min on ice. Cells were then washed in permeabilization buffer, and once more in FACS buffer. Stained cells were resuspended in FACS and acquired on the same day of preparation on a 5-laser Aurora spectral flow cytometer (Cytek).

**Processing of FCS files.** Raw Flow Cytometry Standard (FCS) files were unmixed using the Cytek Spectraflow software. Serotype-specific B cells (live singlet SSC$^{low}$CD3$^-$CD56$^-$CD19$^+$HLADR$^+$PS-SA$^+$ (Spn) or SSC$^{low}$CD7$^-$CD19$^+$HLADR$^+$PS-SA$^+$ (GBS) were gated and exported per PS-specific gate using FlowJo version 10.8 (Supplementary Figs. 1k and 5a), after compensation of spectral overlap. The PS-gated FCS files were then loaded into RStudio version 2021.09.1 to assess cross-reactive cells and to downsample PS-negative cells to 10,000 cells/sample for subsequent elbow clustering. The exported comma-separated values files containing marker expression and antigen-specificity per cell were uploaded to OMIQ. Batch correction of the phenotypic markers per measurement day was performed with Cytonorm ($k = 3$ for Spn dataset, $k = 7$ for GBS dataset), and results were assessed visually. To

analyse high-dimensional data, UMAP[55] analysis and FlowSOM[56] elbow clustering were performed using all parameters from (Supplementary Figs. 3a and 6).

**Statistics and reproducibility.** Two-sided tests were performed (unless indicated otherwise) and non-parametric tests were used if datasets were not normally distributed. Statistical significance for correlation was assessed with Pearson correlations. Single comparisons between groups were assessed by Mann–Whitney or Wilcoxon tests, in case of paired and non-paired comparisons. Multiple-group comparisons were made with one-way ANOVA.

To establish the statistical significance of the phenotypic clusters of the PS-specific B cells, linear regression models with wild bootstrap simulation (9999 resamples) were performed[57]. Wild bootstrapping was employed to account for the fact that model residuals showed heteroscedasticity. The following models were specified: Spn cluster frequency against serotype and vaccination status (including only main effects), for the effect of vaccination or serotype on phenotype. One-sided confidence intervals (CIs) were generated to assess enriched phenotypes only and not depleted phenotypes, given the compositional nature of the data. We compared for each serotype the estimate against the grand mean of all serotypes, and also compared 16 weeks after vaccination versus no vaccination. GBS cluster and isotype frequency were tested against serotype and geographical region (South Africa and the Netherlands) including interaction terms, to determine the effect of geography and serotype on phenotype or isotype. Two-sided CIs were generated to assess the differences of serotype and isotype effects between countries. Statistical comparisons were made through contrasts of regression parameters. Finally, the statistical significance of the contrasts was again assessed through wild bootstrap CIs calculated based on Efron's percentile method with confidence levels adjusted for multiple comparisons according to Bonferroni's method to attain a minimum of 5% confidence[58,59]. As a criterion for data inclusion in regression models, a minimum of five cells per serotype and donor was employed as a cutoff.

**Figures.** Figures were generated with RStudio version 2021.09.1, GraphPad Prism version 9.3.1 and OMIQ. Summary figures were generated with Biorender.

**Reporting summary.** Further information on research design is available in the Nature Portfolio Reporting Summary linked to this article.

## Data availability

The source data behind the figures can be found in Supplementary Data 1. The unmixed spectral flow cytometry FCS files used for the Spn and GBS studies are deposited under restricted access on Zenodo (https://doi.org/10.5281/zenodo.8423267 and https://doi.org/10.5281/zenodo.8425079, respectively) and may be available upon request.

## Code availability

The code used in this study for analysis and the generation of figures is available at https://github.com/HovingD/2023_NatComBiol.

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

## Acknowledgements

PS-specific B cells were provided by AIMM therapeutics. AIMM B cell clone culture was supported by Nienke J. Blomberg and Sanne Reijm (LUMC). Purified GBS capsular PS were provided by Biovac (South Africa). Koen Stam (LUMC) aided with the R scripts for the creation of the figures and analysis. This work was supported by the Bill and Melinda Gates Foundation (INV008088 to S.P.J. and INV021431 to G.K.). The authors gratefully acknowledge the FCF at LUMC, Leiden, the Netherlands (https://www.lumc.nl/research/facilities/fcf) for technical support in the flow cytometry studies.

## Author contributions

D.H. designed and performed the experiments, data analysis and co-wrote the manuscript. A.H.C.M. performed the statistical analysis of the PS-specific B cell clusters. W.H. aided with the high-dimensional analysis. B.A.N., A.C.d.K., B.-R.W. and R.A.M.S. performed the multimer preparations and aided with the spectral flow cytometry experiments. O.R.J.v.H. aided with the analysis and generation of the figures. P.M.v.H. generated the PS-specific B cell clones. D.M.F. and B.C.U. aided with the set-up of the study and provided the PBMC samples from PCV13-vaccinated individuals. N.D. and G.K. provided PBMC samples from South African individuals. C.H.H. aided with the protocol design for the PS modification. S.P.J. designed and performed experiments, data analysis, designed and overviewed the study and co-wrote the manuscript. All authors read and approved the manuscript.

## Competing interests

The authors declare no competing interests.
