## [Peer Review File · Communications Biology]

Combinatorial multimer staining and spectral flow cytometry facilitate quantification and characterization of polysaccharide-specific B cell immunityReviewers' comments:

Reviewer #1 (Remarks to the Author):

This manuscript by Hoving et. al. describes the development and use of a combinatorial multimer-based approach to identify and study polysaccharide-specific B cells, which have been highly understudied in the literature compared to protein-specific B cells. Polysaccharide-specific B cells have been understudied due to a combination of the lack of tools, as well as the diversity of pathogen-derived polysaccharides, two issues this work aims to overcome. Overall, the authors succeed in their goals and make an important contribution to the field by using dual labeling with different PS tetramers bound to different fluorochrome combinations. Which they subsequently use to phenotype cells using flow cytometry with and without vaccination.

A few minor concerns:

Since these studies used a molar ratio of 4:1 PS:SA, it's unclear why "multimer" was used instead of the more conventionally used term of "tetramer", which has been used to describe 4:1 ligand:SA reagents since the first use to detect antigen-specific T cells (PMID 8810254), and B cells (PMID 12505722) not to mention the hundreds of studies since. "Multimer" could mean dimers, trimers, pentamers, etc, and these different types of multimers could yield differences in the phenotype and frequency of cells that bind them.

Fig. S1i. It looks as if there is a slight reduction in binding to individual multimers when dual tetramer staining is used. This may suggest that B cells binding low levels of tetramer may be lost when dual labeling is employed. This does not eliminate the usefulness of this approach but would be helpful if the authors sensitively assessed this and/or discussed how this may alter results compared to studies using single staining.

Fig S2d, e. It is intriguing that multimers and IgM may compete with each other, but it's unclear if the differences presented represent true differences due to low n. Two technical replicates are not enough to draw statistical conclusions here. Three biological replicates are typically the minimum for this type of analysis so it either needs to be repeated or removed from the study as the data presented is inconclusive but important.

Figure 5: Cluster 24: What are the Ig-negative cells and should these truly be considered a unique cluster? Is this just poor resolution of certain isotypes? IgG and IgM overall look pretty dim in Fig S3, suggesting that this panel may be suboptimal. Or is this surface BCR downregulation? Seems odd that one could detect tetramer multimer binding, but not isotype.

It would also be useful to show whether or not fluorochrome choice alters the frequency of detected cells by testing every combination utilized. For example, the cells in the PS9V gate in Fig 2b/3a appear to bind a low level of both PS9V tetramers, while the PS7F-binding cells appear to have both a low binding and high binding population. Is this a technical issue, where the resolution of PS9V-binding cells is lower thereby losing some of the population, or something biologically relevant related to these different PS? This could be a major limitation of the approach that could skew conclusions or result in discrepancies between studies.

Concentrations for PS and multimers would be more useful in molar instead of weight.

The differences in the south African and Dutch cohorts mentioned in the discussion undermines the conclusions drawn in the results and undermines statistical analyses presented in the figure. I think it may be more appropriate to mention this limitation in the results section, and maybe also the figure legend.

Reviewer #2 (Remarks to the Author):

In this study, Dennis Hoving et al. developed and validated a high-throughput method for profiling antigen-specific B cells using combinatorial staining of fluorescently-labeled capsular polysaccharide polymers and observed non-class exchange (IgM+) The link between memory B cells and vaccine-inefficient *Streptococcus pneumoniae* serotypes 1 and 3. Furthermore, they found increased B-cell activation in donors from South Africa, where there was a high incidence of invasive *S. agalactiae* disease, compared with European donors. While this is a potentially interesting study, there are some issues that need to be addressed.

1. Compared with the ELISA method, flow cytometry has higher requirements on equipment and operators, and the detection range is difficult to control. How would you emphasize the usefulness of this study?
2. In this study, the proportion of positive cell population is low. How to ensure the repeatability of the experiment and set the detection interval?
3. The number of patients enrolled is small, and it is recommended to increase the sample size.
4. It is recommended to use this method to test Asian samples, and the accuracy of the results has been verified.
5. Figure 1i: PS7F-SA-BV650 SA-BV650, the proportion of double positive cell population (Q2) is higher than 0.042. I hope you can confirm if the ordinate setting gate should be lower.

Reviewer #3 (Remarks to the Author):

Summary: this manuscript proposes a novel polysaccharide-multimer reagent as a tool to study B cell responses post-vaccination, a sorely limiting hurdle to vaccinology research. The authors demonstrate development of both *Streptococcus pneumoniae* capsular polysaccharide multimer set and *Streptococcus agalactiae* capsular polysaccharide multimer set against donor-derived PBMCs via cytometry.

Overall impression: complex techniques applied to an interesting scientific question. The method is rather detailed in the application of multimers, although it will depend quite heavily on the quality of the PS used.

Specific comments:

- (1) In the second paragraph of the introduction, it's worth noting that the GBS mortality occurs despite WHO recommendations for GBS colonization screening of pregnant people and antibiotic prophylaxis - so vaccination can grant greater protection.
- (2) For the PCV13-vaccinated donor: was there no way to collect a serum sample to perform serologies to determine vaccination status? It may be a simple confirmation test that lends greater credibility for future studies.
- (3) Line 141: "quality control steps" are mentioned, but no cut-offs for unacceptable tests. Were there any quality metrics determined for these quality control steps?
- (4) Line 184: "Over 90% of the cells were generally negative in the other fluorochrome channels". How is "generally negative" defined?
- (5) Line 282-285: cross-reactivity noted between GBS PS of different serotypes - may not just be contaminants, but also shared epitopes across the GBS PS fragments used. More details on the source of the GBS PS may help: what was the purification method, what were approximate sizes of each PS (given they are repeating subunits)?

We thank the Reviewers for all their insightful comments, which helped us to improve the quality of the manuscript. A point-by-point reply to each of the comments raised is provided below.

In addition, we included additional changes to improve clarity and interpretation:

- Fig. 1a, Added staining protocol steps to the workflow.
- Fig. 3e/5a. Renumbered the Log₁₀ axis conversion, ie instead of -3 to -1 the axis now go from 0.001 to 0.1.
- We added to the introduction the recently published NEJM paper describing the phase 2 trial with an investigational GBS conjugate vaccine (Madhi, S.A. Potential for Maternally Administered Vaccine for Infant Group B Streptococcus. N Engl J Med 389, 215-227 (2023))

Reviewer #1 (Remarks to the Author):

This manuscript by Hoving et. al. describes the development and use of a combinatorial multimer-based approach to identify and study polysaccharide-specific B cells, which have been highly understudied in the literature compared to protein-specific B cells. Polysaccharide-specific B cells have been understudied due to a combination of the lack of tools, as well as the diversity of pathogen-derived polysaccharides, two issues this work aims to overcome. Overall, the authors succeed in their goals and make an important contribution to the field by using dual labeling with different PS tetramers bound to different fluorochrome combinations. Which they subsequently use to phenotype cells using using flow cytometry with and without vaccination.

A few minor concerns:

Since these studies used a molar ratio of 4:1 PS:SA, it's unclear why "multimer" was used instead of the more conventionally used term of "tetramer", which has been used to describe 4:1 ligand:SA reagents since the first use to detect antigen-specific T cells (PMID 8810254), and B cells (PMID 12505722) not to mention the hundreds of studies since. "Multimer" could mean dimers, trimers, pentamers, etc, and these different types of multimers could yield differences in the phenotype and frequency of cells that bind them.

We decided to use the term multimer as we have not performed specific experiments, beyond the SA:PS ratio titrations, to confirm with absolute certainty that four units of PS are bound to one single streptavidin molecule. Thus, while on average we have a 4:1 ratio, there might also be some streptavidin molecules with more or less PS molecules.

Fig. S1i. It looks as if there is a slight reduction in binding to individual multimers when dual tetramer staining is used. This may suggest that B cells binding low levels of tetramer may be lost when dual labeling is employed. This does not eliminate the usefulness of this approach but would be helpful if the authors sensitively assessed this and/or discussed how this may alter results compared to studies using single staining.

We thank the reviewer for their excellent observation. Indeed the downside of double staining is the reduction of signal intensity as cellular BCR availability is half, for which we may potentially lose some low BCR-expressing B cells. This is now mentioned in the designated results section of the manuscript (Lines 184-187).

Fig S2d, e. It is intriguing that multimers and IgM may compete with each other, but it's unclear if the differences presented represent true differences due to low n. Two technical replicates are not enough to draw statistical conclusions here. Three biological replicates are typically the minimum for this type of analysis so it either needs to be repeated or removed from the study as the data presented is inconclusive but important.

We agree with the reviewer and have therefore decided to remove these results from the manuscript.

Figure 5: Cluster 24: What are the Ig-negative cells and should these truly be considered a unique cluster? Is this just poor resolution of certain isotypes? IgG and IgM overall look pretty dim in Fig S3, suggesting that this panel may be suboptimal. Or is this surface BCR downregulation? Seems odd that one could detect tetramer multimer binding, but not isotype.

We now added flow cytometry density plots showing the isotype signal intensity and boxplots representing the MFI per cluster in Supplementary Figure 5i and 5j. Average MFIs of cluster 24 suggest that this cluster has very low expression of IgG. We now addressed this in the main body text as well (Line 407). The marker expression of this cluster indicates that these cells are activated (CD95+), and indeed the BCR is known to be downregulated from the surface upon B cell activation (<https://www.ncbi.nlm.nih.gov/pmc/articles/PMC1087239/>). Thus, we agree with the Reviewer's suggestion that this low expression is probably due to internalization. One potential explanation for the fact that we can detect multimer binding but not isotype is that the use of tetramers increases the affinity by recruiting more BCRs than single isotype antibodies, leading to increased signal intensity.

It would also be useful to show whether or not fluorochrome choice alters the frequency of detected cells by testing every combination utilized. For example, the cells in the PS9V gate in Fig 2b/3a appear to bind a low level of both PS9V tetramers, while the PS7F-binding cells appear to have both a low binding and high binding population. Is this a technical issue, where the resolution of PS9V-binding cells is lower thereby losing some of the population, or something biologically relevant related to these different PS? This could be a major limitation of the approach that could skew conclusions or result in discrepancies between studies.

We agree with the reviewer and now included an experiment where polysaccharides from three different serotypes were multimerized with all six streptavidin fluorochromes. We stained three donors each with three different fluorochrome combinations and observed comparable frequencies

for each condition. The results demonstrate that the frequencies detected are not dependent on the fluorochrome combination that is used. Moreover, these data suggest that also the high versus low binding is related to the different PS and not to the fluorochromes, even though signal intensity of course varies as different fluorochromes have a different brightness. These results have been added to supplementary figure 2c and d. We intend to explore this interesting observation further to see if we can link this to B cell or antibody properties in future studies where we are including larger sample sizes longitudinally followed during vaccination and colonization.

Concentrations for PS and multimers would be more useful in molar instead of weight.

We decided to express the concentrations in weight/mL, as the molecular weight varies for each PS, meaning that for the same weight each would have a distinct molarity. While we could have standardized the concentrations in molar rather than weight, we decided to not do this for various reasons. First, the PS are purchased with concentrations provided in weight, as was the manuscript by Zhang et al (ref 32 in the manuscript, Proc Natl Acad Sci U S A 110, 13564-13569 (2013), which first described the biotinylation protocol for the PS, which we followed here to generate the biotinylated PS, making it for us the more logical descriptive parameter. Of note, B cell staining protocols with labelled proteins are also often describing weight rather than molar concentrations.

The differences in the south African and Dutch cohorts mentioned in the discussion undermines the conclusions drawn in the results and undermines statistical analyses presented in the figure. I think it may be more appropriate to mention this limitation in the results section, and maybe also the figure legend.

We agree with the reviewer and have now added this information to the results section (line 387) and the figure legend.

Reviewer #2 (Remarks to the Author):

In this study, Dennis Hoving et al. developed and validated a high-throughput method for profiling antigen-specific B cells using combinatorial staining of fluorescently-labeled capsular polysaccharide polymers and observed non-class exchange (IgM+) The link between memory B cells and vaccine-inefficient *Streptococcus pneumoniae* serotypes 1 and 3. Furthermore, they found increased B-cell activation in donors from South Africa, where there was a high incidence of invasive *S. agalactiae* disease, compared with European donors. While this is a potentially interesting study, there are some issues that need to be addressed.

1. Compared with the ELISA method, flow cytometry has higher requirements on equipment and operators, and the detection range is difficult to control. How would you emphasize the usefulness of this study?

We now highlighted the demands of costs and expertise of flow cytometry approaches in the introduction and have clarified that these methods can overcome the limitations of ELISpot, such as lack high-throughput potential, indirect quantification and inability to perform phenotypic assessment (lines: 94-98).

2. In this study, the proportion of positive cell population is low. How to ensure the repeatability of the experiment and set the detection interval?

We agree with the reviewer that the frequencies of the populations of interests are low, which is to be expected. However, we found that the majority of the populations are above the lower limit of detection (1/100.000 B cells). We established this limit by assessing the frequency of signal detected in our empty fluorochrome combination (for Spn BV711/BUV661 and for GBS BUV615/BUV661). In Figure 3b and Supplementary figures 2e, we show that comparable results are obtained when samples are analysed with both technical duplicates and multiple independent experimental repeats, demonstrating repeatability.

3. The number of patients enrolled is small, and it is recommended to increase the sample size.

We agree with the reviewer that the donor sample size is small. However, we believe that with the extensive quality controls, validations using B cell clones and PBMCs, experimental repeats and statistical analysis approaches we have accounted for these limitations to ensure reproducible results and generate conclusions that are supported by the data. Increasing sample sizes in different cohorts is something we aim to do in the future.

4. It is recommended to use this method to test Asian samples, and the accuracy of the results has been verified.

We agree with the reviewer that it would be very interesting to study serotype specific Bmem in samples from an Asian cohort. China, Vietnam and Thailand have not established a routinely pneumococcal vaccine regimen, and Indonesia only in specific regions. And GBS colonization rates are very low in Nepal, based on recent multi-centre studies (personal communication Gaurav Kwatra). It would be very interesting to compare naturally-induced immunity versus vaccine immunization in samples from these countries, while simultaneously assessing potential differences in Bmem frequencies driven by serotype-specific endemicity across countries (ie, in comparison with Europe). Unfortunately, we do not have access to such sample cohorts as of now. For GBS we have included samples from European and African donors and have seen that high-endemic areas show Bcells with increased activated and class-switched phenotype.

5. Figure 1i: PS7F-SA-BV650 SA-BV650, the proportion of double positive cell population (Q2) is higher than 0.042. I hope you can confirm if the ordinate setting gate should be lower.

In the top plot, which the Reviewer is alluding to, there was no SA-BV650 added. We get similar signals with unstained cells. The reason that the negative population goes so high (up to 10^4 for both BV650 and BV785) is that these B cell clones are extremely autofluorescent. We have now changed the labels of the panels to clarify the conditions.

Reviewer #3 (Remarks to the Author):

Summary: this manuscript proposes a novel polysaccharide-multimer reagent as a tool to study B cell responses post-vaccination, a sorely limiting hurdle to vaccinology research. The authors demonstrate development of both *Streptococcus pneumoniae* capsular polysaccharide multimer set and *Streptococcus agalactiae* capsular polysaccharide multimer set against donor-derived PBMCs via cytometry.

Overall impression: complex techniques applied to an interesting scientific question. The method is rather detailed in the application of multimers, although it will depend quite heavily on the quality of the PS used.

Specific comments:

(1) In the second paragraph of the introduction, it's worth noting that the GBS mortality occurs despite WHO recommendations for GBS colonization screening of pregnant people and antibiotic prophylaxis - so vaccination can grant greater protection.

We thank the reviewer for the suggestion, this clarification is now added to the introduction of the manuscript (Line: 64-65).

(2) For the PCV13-vaccinated donor: was there no way to collect a serum sample to perform serologies to determine vaccination status? It may be a simple confirmation test that lends greater credibility for future studies.

We agree with the reviewer that antibody data would be a great addition to the cellular results. Unfortunately, serum was unavailable from the donors included in this study. However, we are currently working on a new study where this comparison will be included.

(3) Line 141: "quality control steps" are mentioned, but no cut-offs for unacceptable tests. Were there any quality metrics determined for these quality control steps?

We now added further clarification to the validation with the quality control steps in the results section (Lines: 133-136). From our experience, we observe consistently similar values with different

batches. Moreover, we foresee that these absolute values (either OD of the biotinylation ELISA, inhibition percentage of competition ELISA and MFI of serum-coated beads) may be personal, equipment and reagent-specific. Therefore it was impossible for us to define a specific cut-off below which the prepared reagents are likely to fail, apart from being different from the negative controls included in these QC steps. Because of this, we assessed the quality control steps qualitatively, and not quantitatively, including the adequate controls in each step.

(4) Line 184: "Over 90% of the cells were generally negative in the other fluorochrome channels". How is "generally negative" defined?

We have now assessed the percentage of cross-reactive cells over all donors included in the cross-sectional study, observing that a median of 88.7% (83.5-97.2%) of PS-positive cells are serotype-specific, and give signal only for two out of the six channels, with the remainder being cross-reactive. This figure is placed in Supplementary figure 2e, and corrected in the text (Lines 245-247).

(5) Line 282-285: cross-reactivity noted between GBS PS of different serotypes - may not just be contaminants, but also shared epitopes across the GBS PS fragments used. More details on the source of the GBS PS may help: what was the purification method, what were approximate sizes of each PS (given they are repeating subunits)?

We thank the reviewer for their observation, and agree that the quality of the PS used is key for having good quality data. The observation that blocking Siglecs substantially reduced cross-reactivity (supplementary figures 5b and c) indicates that at least part of the cross-reactivity was due to the presence of the sialic acids on the repeating subunits of the GBS PS. However, there are indeed also other potential sources of cross-reactivity, such as contaminants and shared epitopes. The GBS PS provided by Biovacc is clinical-grade material used as the basis for vaccines that are being tested in clinical trials, and is thus of excellent quality. We now added the information on the GBS PS purification and sizes in the methods section of the manuscript (Lines 492-495).

REVIEWERS' COMMENTS:

Reviewer #1 (Remarks to the Author):

Thank you for carefully addressing my concerns. I am satisfied by your responses and edits and have no other concerns. This will be a nice addition to the field and will hopefully be utilized by others.